# Mesenchymal Stromal Cells Regulate Sialylations of N-Glycans, Affecting Cell Migration and Survival

**DOI:** 10.3390/ijms22136868

**Published:** 2021-06-26

**Authors:** Kayla Templeton, Meiby Ramos, Jacqueline Rose, Bryan Le, Qingwen Zhou, Amin Cressman, Stephanie Ferreyra, Carlito B. Lebrilla, Fernando Antonio Fierro

**Affiliations:** 1Stem Cell Program, Institute for Regenerative Cures, University of California Davis, Sacramento, CA 95817, USA; kmtempleton@ucdavis.edu (K.T.); meibyramos@csus.edu (M.R.); jr2275@humboldt.edu (J.R.); bynle@ucdavis.edu (B.L.); Amin.Cressman@humboldt.edu (A.C.); stephanieferreyra@csus.edu (S.F.); 2Department of Chemistry, University of California Davis, Davis, CA 95616, USA; qwzzhou@ucdavis.edu (Q.Z.); cblebrilla@ucdavis.edu (C.B.L.); 3Department of Cell Biology and Human Anatomy, Davis, CA 95616, USA

**Keywords:** mesenchymal stromal cells, glycosylation, sialic acid, sialyltransferases, migration, survival

## Abstract

N-Glycosylations are an important post-translational modification of proteins that can significantly impact cell function. Terminal sialic acid in hybrid or complex N-glycans has been shown to be relevant in various types of cancer, but its role in non-malignant cells remains poorly understood. We have previously shown that the motility of human bone marrow derived mesenchymal stromal cells (MSCs) can be modified by altering N-glycoforms. The goal of this study was to determine the role of sialylated N-glycans in MSCs. Here, we show that IFN-gamma or exposure to culture media low in fetal bovine serum (FBS) increases sialylated N-glycans, while PDGF-BB reduces them. These stimuli alter mRNA levels of sialyltransferases such as ST3Gal1, ST6Gal1, or ST3Gal4, suggesting that sialylation of N-glycans is regulated by transcriptional control of sialyltransferases. We next show that 2,4,7,8,9-pentaacetyl-3Fax-Neu5Ac-CO2Me (3F-Neu5Ac) effectively inhibits sialylations in MSCs. Supplementation with 3F-Neu5Ac increases adhesion and migration of MSCs, as assessed by both videomicroscopy and wound/scratch assays. Interestingly, pre-treatment with 3F-Neu5Ac also increases the survival of MSCs in an in vitro ischemia model. We also show that pre-treatment or continuous treatment with 3F-Neu5Ac inhibits both osteogenic and adipogenic differentiation of MSCs. Finally, secretion of key trophic factors by MSCs is variably affected upon exposure to 3F-Neu5Ac. Altogether, our experiments suggest that sialylation of N-glycans is tightly regulated in response to environmental cues and that glycoengineering MSCs to reduce sialylated N-glycans could be beneficial to increase both cell migration and survival, which may positively impact the therapeutic potential of the cells.

## 1. Introduction

Mesenchymal Stromal Cells (MSCs) have been widely recognized as a key cell type in regenerative medicine [1]. Over one thousand clinical trials have tested MSCs as therapeutics to promote wound healing, tissue regeneration, and to reduce inflammation, among others [2]. MSCs have a remarkable safety profile because they can evade the immune system, while most human cells are rarely tolerated in allogeneic settings [3]. Despite this attention, very few clinical trials have met therapeutic expectations and led to commercialization of MSCs [4]. The limited success of MSCs could be due to the use of sub-optimal doses, inadequate routes of administration, or host predisposition [5].

The efficacy of MSCs is also likely hindered by the inability to target MSCs to tissues of interest [6]. For many applications, MSCs require active cell motility to reach their intended sites, but current approaches to improve cell motility have been only modestly successful, with no clear impact on the cell’s therapeutic effect [7]. Improving MSC migration should also result in less invasive delivery methods [8].

In addition, MSCs show poor survival after transplantation, which is also likely limiting the therapeutic effect of the cells [9]. One strategy to improve the retention of MSCs is to genetically modify the cells [10], but such approaches increase the risks for oncogenic transformation. Alternative approaches, such as hypoxic preconditioning or glucose supplementation, increase cell survival without jeopardizing the safety profile of MSCs [11,12], but exert broad effects on the cells, which could negatively impact their therapeutic effect.

Based on recent developments in the detection and modulation of N-linked glycosylations (N-glycans), we propose to harness these post-translational modifications to improve MSC’s motility and survival. Some key advantages are that N-glycoforms can be modified without genetic modification and that these changes are only transient, depending on protein turnover [13].

The study of N-glycans on cell migration has been mostly limited to selectin ligands [14], which are involved in cell tethering and rolling [6,8]. Since MSCs do not express selectins, very little is known about how different N-glycoforms impact migration of MSCs. However, it has been shown that engineering glycosylations increases homing of MSCs to the bone marrow [15]. We have shown that inducing N-glycan core-fucosylations or high mannose N-glycans promote migration of MSCs [13,16]. Sialylations of N-glycans are known to play essential roles in the immune system and cancer [17,18]. However, their exact role varies widely based on the cell type, due to the large differences in expressed glycoproteins. We, therefore, examined how sialylations of N-glycans would affect MSCs on different parameters, including cell migration and survival.

In addition, from a basic cell biology perspective, it remains minimally understood how N-glycan modifications are regulated. A common misconception is that glycosyltransferases and glycosidases of the Golgi Apparatus and the Endoplasmic Reticulum are constitutively expressed to fulfil basic cellular functions [19]. However, a growing trend acknowledges the interconnectedness of signaling pathways, which direct cell fate and function, with traditionally considered “housekeeping” functions, such as cell metabolism and protein homeostasis [20]. In line with this trend, we have shown that basic fibroblast growth factor (bFGF) induces transcriptional increase of fucosyltransferase 8, increases N-glycan core-fucosylations, and promotes cell motility [16]. Based on these previous findings, here we hypothesized that environmental cues would also regulate sialylations of N-glycans in MSCs.

## 2. Results

### 2.1. Signals Regulating Sialylated N-Glycans in MSCs

Based on our previous observation that N-glycan fucosylations were increased in response to bFGF [16], we aimed to identify conditions that would alter sialylations in MSCs. As a surrogate method to determine sialylation levels, we used SNA coupled to a fluorophore for measurements using flow cytometry. SNA is a lectin derived from Elderberry bark (*Sambucus nigra*) that preferentially binds to sialic acid attached to terminal galactose in α-2,6, and to a lesser degree, in α-2,3 linkage [21]. Exposure to the pro-inflammatory cytokine Interferon gamma (IFN-γ) caused a strong increase in SNA binding (Figure 1A,B, and Appendix A), suggesting an increase in sialylated N-glycans. As a potential underlying mechanism, we hypothesized that IFN-γ would increase expression of sialyltransferases. In fact, MSCs exposed to IFN-γ show a significant increase of mRNA levels of ST6 β-galactoside α-2,6-sialyltransferase 1 (ST6Gal1) over time (Figure 1C). IFN-γ also causes a transient increase of ST3 Beta-Galactoside Alpha-2,3-Sialyltransferase 1 (ST3Gal1). However, we also found that IFN-γ caused a transient and small, but significant decrease of ST3 Beta-Galactoside Alpha-2,3-Sialyltransferase 4 (ST3Gal4). IFN-γ has been reported to activate multiple signaling pathways, including the NF-κB pathway, ERK1/2, and PI3K/Akt [22,23]. To test which of these downstream pathways is involved in IFN-γ mediated-induction of sialylations, we pre-treated MSCs for 1 h with specific inhibitors, prior to the 24 h stimulation with IFN-γ. Inhibition of NF-κB with pyrrolidine dithiocarbamate (PDTC; 200 μg/mL) completely abrogated the effect of IFN-γ, while inhibition of ERK1/2 with UO126 (10 μM) did not affect SNA binding (Figure 1D). Surprisingly, inhibition of PI3K/Akt with LY294002 (10 μM) caused a further increase of SNA binding. These results suggest that IFN-γ stimulates sialylations in an NF-kB dependent manner, while inhibition of the Akt pathway leads to increased sialylated N-glycans.

The seeding density of MSCs significantly affects sialylations, although the effect is rather small (Appendix A). Interestingly, exposure of MSCs to culture media low in fetal bovine serum (FBS) also caused an increase in sialylations (Figure 1E,F). In agreement with our observation with IFN-γ, exposure to 0.5% FBS caused a steady increase of ST6Gal1 mRNA levels over time (Figure 1G), while the mRNA levels of ST3Gal1 and ST3Gal4 did not change in response to low FBS. These results suggest that overall stress conditions (i.e., inflammatory cytokines, high cell density, or reduced FBS) cause an increase of sialylated N-glycans in MSCs.

The mitogen platelet derived growth factor beta (PDGF-BB) is known to activate the PI3K/Akt pathway in MSCs [24,25]. Since inhibition of PI3K/Akt caused an increase in SNA-binding, we hypothesized that stimulation with PDGF-BB should cause a decrease in SNA-binding. Indeed, as shown in Figure 1H,I, PDGF-BB (10 ng/mL) downregulated sialylated N-glycans. This observation only partially correlated with the mRNA levels of ST3Gal1, ST6Gal1, and ST3Gal4. ST3Gal1 mRNA was slightly increased after 6 and 12 h with PDGF-BB, while ST3Gal4 mRNA was significantly downregulated after 24 h with PDGF-BB (Figure 1I). Finally, we tested how inhibition of signaling pathways affected PDGF-BB mediated reduction in sialylations. While PDGF-BB reduces SNA binding, pre-treatment with inhibitors against NF-kB, ERK1/2 and PI3K/Akt restored sialylation levels to similar levels found in control cells (Figure 1K). These results suggest that PDGF-BB reduces sialylations in MSCs, while blockade of downstream signaling reverses this effect.

### 2.2. 3F-Neu5Ac Effectively Inhibits Sialylated N-Glycans in MSCs

To test how sialylation of N-glycans affects the biology of MSCs, we evaluated the use of 3F-Neu5Ac, which is a sialyltransferase inhibitor that structurally resembles sialic acid and, due to a fluorine atom, remains strongly attached to the enzymes to effectively block their catalytic activity [26]. Although 3F-Neu5Ac is commonly used at 200 μM [26], we found that in MSCs, 50 μM is just as effective at reducing sialylations (Figure 2A). This concentration significantly reduced sialylations after 1, 2, or 3 days of exposure (Figure 2B). Since the inhibition was slightly higher after 2 days than after 1 day, all further experiments were performed by treating MSCs with 3F-Neu5Ac at 50 μM, for 2 days. To confirm the inhibition of sialylated N-glycans and elucidate how it would impact the abundance of other N-glycoforms, MSCs treated with or without 3F-Neu5Ac were analyzed by nano-flow liquid chromatography/electrospray ionization quadruple time-of-flight mass spectrometry (Nano-LC/ESI QTOF MS). As shown in Figure 2C,D, 3F-Neu5Ac caused a strong reduction of sialofucosylated and sialylated N-glycans. Consequently, neutral and fucosylated N-glycans were significantly increased, while the proportion of high mannose N-glycans was not affected. These results show that treating MSCs for 2 days with 50 μM 3F-Neu5Ac effectively reduces sialylated N-glycans.

### 2.3. 3F-Neu5Ac Promotes Cell Motility in MSCs

Having confirmed the efficacy of 3F-Neu5Ac to reduce N-glycan sialylations in MSCs, we next sought to investigate how a reduction in sialylations would impact different parameters within cells. To investigate the effect of 3F-Neu5Ac on cell migration, we used two types of experiments: wound/scratch assays and videomicroscopy. In the wound/scratch assay, MSCs were plated to confluence with inserts leaving a constant 0.5 mm gap or “wound” in the monolayer. In these assays, MSCs pre-treated with 3F-Neu5Ac consistently showed a mild increase in wound closure compared to controls, which is indicative of increased cell migration (Figure 3A,B). In agreement with these observations, also tracking of individual cells using videomicroscopy shows that MSCs pretreated with 3F-Neu5Ac have increased motility as compared to control MSCs (Figure 3C,D).

Since cell adhesion has a strong impact on cell migration, we also tested if reduced sialylations would impact cell adhesion. This assay was performed with MSCs in suspension, left to attach to uncoated culture plates for 10 min. As shown in Figure 3E, pre-treatment with 3F-Neu5Ac caused a significant increase in cell adhesion.

Since IFN-γ, low FBS, and PDGF-BB were effective modulators of sialylations in MSCs, we tested how these factors would affect cell migration. Pre-treatment with IFN-γ or 0.5% FBS inhibited cell migration, while PDGF-BB had no significant impact, as measured in wound/scratch assays (Appendix A). Cell adhesion was not affected by IFN-γ or PDGF-BB, but pretreating MSCs in 0.5% FBS caused a marked reduction in cell adhesion, as compared to controls (Appendix A). Altogether, these experiments suggest that reducing sialylations promotes migration of MSCs.

### 2.4. Reducing Sialylations Promotes Survival of MSCs in an Ischemia Model In Vitro

In endothelial cells, macrophages, and lymphoblastic cell lines, a reduction in sialylations is associated with increased apoptosis [27,28,29,30]. We, therefore, evaluated if pre-treating MSCs with 3F-Neu5Ac would impact cell survival. For this, we mimic “ischemia” in vitro, by culturing the cells in serum free-media and 1% oxygen, with no further medium changes. These conditions lead to gradual cell death, likely due to nutrient deprivation [11]. Surprisingly, we found that pre-treatment with 3F-Neu5Ac caused an increase in cell survival over time. As shown in Figure 4A, the number of living cells was significantly higher after 9 and 12 days in ischemic culture conditions, when pretreated with 3F-Neu5Ac, as compared to control cells. We also tested if the conditions that were found to modulate sialylations (IFN-γ, 0.5% FBS and PDGF-BB) would affect cell survival. After the 2 days pre-treatment, cells treated with PDGF-BB had increased, and cells treated with low FBS were reduced, as compared to the control (Appendix A). However, after 9 days in ischemic culture conditions, only cells pre-treated with 0.5% FBS showed decreased survival, as compared to control (Appendix A). Therefore, although reducing sialylations promoted cell survival, the signals that regulate sialylations did not necessarily impact cell survival in the same manner. Of note, treating MSCs with 3F-Neu5Ac did not affect cell proliferation (Appendix A). 

To explore how a reduction in sialylation could promote cell survival, an apoptosis protein array was performed with cells after 6 days under ischemic culture conditions, which is the earliest time point where we noticed a trend toward higher survival in MSCs pretreated with 3F-Neu5Ac. Among the 35 proteins detected, only catalase was consistently increased in MSCs derived from two donors (Figure 4B,C). Catalase catalyzes the conversion of hydrogen peroxide (H_2_O_2_) into water (H_2_O) and molecular oxygen (O_2_) and is, therefore, a key enzyme in the reduction of reactive oxygen species (ROS) [31]. The increase in catalase expression was further confirmed at mRNA level (Figure 4D), also tested in cells after 6 days in ischemia. However, the differences were rather modest at this time point, making it uncertain if this would impact overall oxidative stress within the cells. In summary, pre-treating MSCs with 3F-Neu5Ac promotes cell survival in vitro, which might be associated with increased expression of catalase, possibly reducing oxidative stress in the cells.

### 2.5. Reducing Sialylations Affect the Secretion of Trophic Factors and the Differentiation Potential of MSCs

Considering the beneficial effect of 3F-Neu5Ac on cell migration and survival, we next examined if this approach would affect MSCs in other aspects of their biology. Many clinical applications of MSCs rely on the paracrine activity of the cells, mediated by secretion of trophic factors [32]. We, therefore, measured by ELISA the secretion of pro-angiogenic proteins VEGF, Interleukin 8 (IL-8) and Angiopoietin 2, and the scaffold protein Hyaluronan in MSCs treated for 2 days with or without 3F-Neu5Ac. We found that the effect of 3F-Neu5Ac was variable: Angiopoietin 2 was increased, Hyaluronan decreased, and IL-8 and VEGF were not significantly affected (Figure 5A). These results indicate that pre-treating MSCs with 3F-Neu5Ac can affect the secretion of trophic factors in a variable way.

To also assess if 3F-Neu5Ac would affect the osteogenic and adipogenic differentiation of MSCs, we used two approaches. First, cells were only pre-treated with the inhibitor, 2 days prior to the start of differentiation. Second, MSCs were continuously treated with the inhibitor (including the 2 days pre-treatment and addition of 3F-Neu5Ac with every medium change during differentiation). Adipogenesis was measured by Oil-red O staining, which binds to the triglycerides contained in adipocytes, while osteogenesis was measured by quantification of Alizarin Red S (ARS) staining, which binds to mineralized calcium. As shown in Figure 5B,C, both pre-treatment and continuous treatment with 3F-Neu5Ac caused a significant reduction of adipogenic differentiation of the cells. In contrast, osteogenesis was only significantly reduced after continuous treatment with 3F-Neu5Ac.

## 3. Discussion

Perhaps there are two primary conclusions stemming from this report: First, various environmental signals (IFN-γ, low serum supplementation, cell density, and PDGF-BB) affect sialylations of glycoproteins in MSCs. Second, changes in sialylations impact MSCs on multiple levels, including migration and survival.

A common notion is that N-glycans are attached to glycoproteins in a constitutive manner. However, our results suggest that protein function is further regulated by altering the N-glycoforms attached to them. This regulation seems to occur at the transcriptional level of glycosyltransferases and perhaps glycosidases. Based on our current knowledge, these changes in the synthesis of N-glycans should affect many glycoproteins in a nonspecific manner. The concept of N-glycans altering cell function is well established and has proven to be very important in relation to health and disease [17]. Interestingly, the conditions that increase sialylations (the pro-inflammatory cytokine IFN-γ, serum deprivation, high cell density, and inhibition of Akt) are possibly all stress-inducers, although through different mechanisms. This leaves an important open question: does increasing sialylations mitigate or exacerbates cellular stress? We show that pre-treatment of MSCs with 3F-Neu5Ac promotes cell survival in an in vitro ischemia model (possibly by reducing oxidative stress). However, we did not examine the effect of directly increasing sialylated N-glycans, which deserves further experimentation.

Another limitation of this study is that for most assays, sialylation levels were inferred based on SNA binding. Experiments using other sialic acid-binding lectins are necessary to further confirm our conclusions and to elucidate if the sialylations attached through specific linkages (e.g., α2,6 vs. α2,3) have differential effects on the cells. Of note, our mass spectrometry results suggest that the inhibitor 3F-Neu5Ac strongly reduces all sialylations, regardless of the type of linkage. Still, a largely unexplored area is how the different types of linkages of sialic acid impact glycoprotein function. Such studies will require the modulation of individual sialyltransferases. In addition, it is possible that sialylations are also regulated by levels of sialidases (neuraminidases) [18], which also requires further investigation.

Reducing sialylations with 3F-Neu5Ac promoted cell migration in vitro, as observed in wound/scratch assays and videomicroscopy. In line with this observation, treating MSCs with IFN-γ or 0.5% FBS (which increase sialylations) inhibited cell migration. However, PDGF-BB (which reduces sialylations) did not impact cell migration. It is important to remember that environmental cues trigger a plethora of biochemical changes that ultimately affect cell function in a complex manner. The modification of N-glycoforms is only one layer of cellular regulation.

We recently showed that inducing high mannose N-glycans using Kifunensine also promotes migration of MSCs [13] and cancer cells [33]. Kifunensine also affected the secretion of angiogenic factors, osteogenesis, and adipogenesis in a very similar fashion to 3F-Neu5Ac [13]. Since high-mannose N-glycans are not sialylated, perhaps both Kifunensine and 3F-Neu5Ac affect the cells through a common mechanism by reducing terminal sialic acid. Still, the underlying mechanisms for how reduced sialylations promote cell migration, affect secretion of angiogenic proteins, and inhibit differentiation remains unknown.

Our results suggest that, depending on the intended clinical application of MSCs, pre-treating the cells with 3F-Neu5Ac may positively impact the therapeutic outcome by promoting cell migration and survival. Future in vivo studies are clearly pending to draw such conclusions. Of note, IFN-γ is commonly used to “prime” or “activate” MSCs toward a stronger anti-inflammatory phenotype [34,35]. In this context, it will be particularly relevant to combine IFN-γ and 3F-Neu5Ac pre-treatments, to enhance the therapeutic effect of MSCs.

One remarkable finding reported here is that MSCs pre-treated with 3F-Neu5Ac survived at higher numbers in our in vitro ischemia model. Future experiments should examine survival at even longer time points. Additionally, the underlying mechanism remains unresolved. MSCs pre-treated with 3F-Neu5Ac show a mild increase of catalase levels at day 6. Does this difference in catalase levels change over time? Does this difference impact overall oxidative stress? Is this possible reduction of oxidative stress associated with increased cell survival? Finally, the effect of 3F-Neu5Ac is expected to be transitory and dependent on protein turnover, but the exact time for how long sialylations remain reduced in response to this inhibitor, remains to be studied.

To our knowledge, this is a first report showing regulation of sialylated N-glycans and examination of the role of sialylated N-glycans in MSCs. Our conclusions are quite contrary to observations in cancer cells (where often increased sialylation are associated with metastasis and survival to specific drugs [29,30,36,37]), highlighting how sialylated N-glycans affect cells in a cell type-specific manner, which conversely is likely associated to the expression of different glycoproteins in cells [17]. Finally, our results further support the notion of glycoengineering MSCs, to safely improve their therapeutic efficacy, by either promoting cell survival or increasing their cell motility.

## 4. Materials and Methods

### 4.1. MSC Isolation and Expansion

MSCs were isolated from commercially available human bone marrow (StemExpress, Sacramento, CA, USA). Bone marrow aspirates were passed through 90 μm pore strainer for isolation of bone spicules. Then, bone marrow aspirates were diluted with an equal volume of phosphate-buffered saline (PBS) and centrifuged over Ficoll at 700 g for 30 min. Mononuclear cells from the Buffy Coat and bone spicules (if any) were plated in plastic culture flasks, using Minimum Essential Medium α (MEMα-HyClone, Logan, UT, USA) plus 10% Fetal Bovine Serum (FBS-Phoenix Scientific, San Marcos, CA, USA) (standard culture media). After two days, non-adherent cells were removed by washing 2–3 times with PBS. MSCs between passage 3–6 were used for experimentation to ensure pure MSC cultures. Additionally, higher passage cells may become senescent, which lead to alteration of gene expression, arrest in cell proliferation, and resistance to apoptosis. 

### 4.2. SNA Binding

To measure sialylations, we used SNA (Vector Labs, Burlingame, CA, USA), a lectin isolated from the bark of elderberry (*Sambucus nigra*), which binds to sialic acid that is terminally attached to galactose in an α2,6 linkage and to a lesser degree to α2,3 linkage. For sialylation analysis using 3F-Neu5Ac (Thermo Fisher Scientific, Waltham, MA, USA), MSCs were initially treated with varying concentrations (0, 50, 100, and 200 μM) and for various time points (1, 2, and 3 days). Subsequently, for all other experiments with 3F-Neu5Ac, MSCs were treated with 50 μM for 2 days. For sialic acid expression analysis by IFN-γ and PDGF-BB and pathway inhibitors (PDTC- 200 μg/mL, UO126-10 μM, and LY294002- 10 μM), MSCs were first treated with the inhibitors and incubated for one hour at 37 °C in 20% O_2_. Then, IFN-γ (50 ng/mL) or PDGF-BB (10 ng/mL) was added (without a medium change) and cells were incubated for an additional 24 h, prior to analyzing them using flow cytometry. MSCs were then lifted using trypsin and centrifuged at 500× *g* for 5 min. Cells were then washed with PBS, followed by another centrifugation at 500× *g* for 5 min. The supernatant was removed and MSCs were incubated with SNA lectin (diluted 1:50 in PBS), which was conjugated with FITC-labeled ligands for 30 min at 4 °C. Then, MSCs were washed with PBS and measured using Attune NxT flow cytometer. All inhibitors and recombinant proteins were a kind gift by Dr. Ping Zhou, at UC Davis.

### 4.3. Gene Expression Analysis

To measure gene expression, total RNA was extracted from MSCs using Direct-zol RNA Miniprep kit (Zymo Research, Irvine, CA, USA), following manufacturer’s instructions. RNA was reverse transcribed using Taqman reverse transcription reagents (Thermo Fisher). mRNA levels of sialyltransferases and catalase were measured using TaqMan gene expression assays (Invitrogen, Carlsbad, CA, USA) and TaqMan Universal Master Mix reagents (Invitrogen). Primers and probe can be identified by the following Assay ID: GAPDH: Hs03929097_g1; ST3Gal1: Hs00161688_m1; ST6Gal1: Hs00949382_m1; ST3Gal4: Hs00920870_m1; Catalase: Hs00156308_m1.

### 4.4. N-Glycan Profile Analysis

As in previous studies [13,16], one million MSCs treated for different time points with or without 3F-Neu5Ac (50 μM, for 48 h) were lifted with trypsin, washed once with PBS, and resuspended in homogenization buffer (0.25 M sucrose, 20 mM HEPES-KOH (pH 7.4) and 1:100 protease inhibitor mixture (EMD Millipore, Burlington, MA, USA)). Cells were then lysed on ice using a probe sonicator, and lysates were pelleted by centrifugation at 2000× *g* for 10 min to remove the nuclear fraction and cells that did not lyse, followed by a series of ultracentrifugation steps at 200,000× *g* for 45 min to remove other non-membrane subcellular fractions [38]. Membrane pellets were then suspended in 100 μL of 100 mM NH_4_HCO_3_ in 5 mM dithiothreitol and heated for 10 s at 100 °C to thermally denature the proteins. To release the glycans, 2 μL of peptide N-glycosidase F (New England Biolabs, Ipswich, MA, USA) was added to the samples, which were then incubated at 37 °C in a microwave reactor for 10 min at 20 watts. After the addition of 400 μL of ice-cold ethanol, samples were frozen for 1 h at −80 °C to precipitate deglycosylated proteins and centrifuged for 20 min at 21,130× *g*. The supernatant containing N-glycans was collected and dried. 

Released N-glycans were purified by solid-phase extraction using porous graphitized carbon packed cartridges (Grace, Deerfield, IL, USA). Cartridges were first equilibrated with alternating washes of nanopure water and a solution of 80% (*v*/*v*) acetonitrile and 0.05% (*v*/*v*) trifluoroacetic acid in water. Samples were loaded onto the cartridge and washed with nanopure water at a flow rate of 1 mL/min to remove salts and buffer. N-Glycans were eluted with a solution of 40% (*v*/*v*) acetonitrile and 0.05% (*v*/*v*) trifluoroacetic acid in water and dried. The analysis was performed using nanoflow liquid chromatography/electrospray ionization quadrupole time-of-flight mass spectrometry (NanoLC/ESI-QTOF-MS) as previously described [38].

### 4.5. In Vitro Migration Assays

For wound/scratch assays, cells were seeded into 24 well plates, containing cytosolic inserts (Cell Biolabs, San Diego, CA, USA), at 25,000 cell/500 μL per half. To test 3F-Neu5Ac, MSCs were plated into inserts in standard media, with no supplement (control) or 3F-Neu5Ac (50 μM) and incubated for 48 h. To test conditions that alter sialylations, MSCs were plated into inserts in standard media, with no supplement (control), IFN-γ (50 ng/mL), PDGF-BB (10 ng/mL) or low serum (0.5% FBS) incubated for 24 h. Following incubation, inserts were lifted, leaving an open area. Images were taken of the area immediately after insert removal and 24 h later. TScratch software was utilized to quantify the percentage of wound closure. 

For videomicroscopy, cells were plated onto 35 mm Petri dishes (5000 cells/dish), incubated as described, and placed in a Biostation microscope (Nikon, Tokyo, Japan), while maintaining the cells at 37 °C and 5% CO_2_. During recording, each dish was photographed every 5 min in 10 fields of view for over 20 h. Movies were analyzed using the plugin MTrack from ImageJ software to determine individual cell displacement over time (speed). At least 20 cells per MSC-donor per condition were analyzed.

### 4.6. Adhesion Assay

MSCs were cultured for 48 h in standard culture media, with no supplement (control), 3F-Neu5Ac (50 μM), IFN-γ (50 ng/mL), PDGF-BB (10 ng/mL) or low serum (0.5% FBS). Cells were then lifted using trypsin, washed once with PBS and reseeded for 10 min in a 12-well plate (40,000 cells/well in 4 replicates) over a rocking platform. Then, nonadherent cells are discarded, plates were washed once with PBS, and remaining cells (i.e., attached cells) counted using trypan blue exclusion dye and hemocytometer.

### 4.7. In Vitro Cell Survival Assay

To determine MSC survival, an established in vitro ischemic model was used, where cells were incubated in hypoxia (1% O_2_) and serum-free media, with no additional medium changes. First, MSCs were seeded at 10,000 cells per cm^2^ in 12-well plates in MEMα plus FBS. After 24 h, the media was changed to MEM α + 10%FBS with or without 3F-Neu5Ac (50 μM), as pre-treatment. After 48 h (Day 0), cells received a medium change to serum-free media (MEM α only) with or without 3F-Neu5Ac. Cells were counted at days 0, 6, 9, and 12, by lifting the cells using trypsin and counting them using trypan blue exclusion dye and hemocytometer.

For cell survival analysis with IFNγ, PDGF-BB, and 0.5% serum (low serum), MSCs were seeded at 10,000 cells per cm^2^ in 12-well plates in MEMα + 10% FBS. After 24 h, MSC pre-treatment was MEMα + 10% FBS with or without IFN-γ (50 ng/mL) or PDGF-BB (10 ng/mL). For low serum pre-treatment, MSCs received MEMα + 0.5% FBS. After 48 h (Day 0), MSCs received a medium change to MEMα only +/-IFNγ or PDGF-BB and were incubated in 1% O_2_ to simulate hypoxic conditions during bone fracture repair. MSCs were counted at days 0 and 9, by lifting the cells using trypsin and counting them using Trypan blue exclusion day and hemocytometer.

### 4.8. Apoptosis-Proteome Array

To analyze the difference in apoptotic pathways, MSCs were seeded at 10,000 cells per cm^2^ in T75 culture flasks. After 24 h, MSCs receive pre-treatment +/-3F-Neu5Ac in MEM α + 10%FBS. After 48 h, media was changed to MEM α only +/-3FNeu5Ac. After 6 days, MSCs were detached using trypsin and resuspended in RIPA buffer (Thermo Fisher) containing 1% Halt Proteinase and Phosphatase inhibitors (Thermo Fisher) to prevent protein degradation. For maximum efficiency of protein extraction, strong agitation was applied for 20 min on ice. Samples were centrifuged at maximum speed for 15 min at 4 °C and supernatant was transferred to clean Eppendorf tube and kept at −80 °C. Extracted proteins were analyzed using Human Apoptosis Antibody Array (R&D Systems, Minneapolis, MN, USA), a test used to detect the expression of 35 apoptosis related proteins from a single sample. To analyze the results, pixel intensity of each detected protein was measured using Adobe Photoshop. 

### 4.9. Cell Proliferation Assay

To determine the effects on sialylation inhibition using 3F-Neu5Ac on cell proliferation, MSCs were seeded at 2500 cells per cm^2^ in 12-well plates. After 24 h (Day 0), cells were treated with or without 3F-Neu5Ac in MEMα + 10% FBS. Every two days, MSCs were detached using trypsin and counted using Trypan blue exclusion dye and hemocytometer. MSCs received a media change on day 3, to account for the usage of nutrients. 

### 4.10. Analysis of Secreted Proteins

MSCs were plated at 10,000 cells/cm^2^ in 12-well plates in triplicate. The day after, medium was changed to standard culture media with or without 3F-Neu5Ac (50 μM). After 48 h, supernatants were stored at -80 °C. Secretion of hyaluronan, Angiopoietin-2, VEGF and IL-8 was measured by enzyme-linked immunosorbent assay (ELISA) using the respective DuoSet kits (R&D Systems), following the manufacturer’s instructions.

### 4.11. Differentiation Assays

Osteogenic and adipogenic differentiation were performed as previously described [13], with an initial cell density of 10,000 MSCs/cm^2^ with regular medium changes every 3–4 days. For both types of assays, cells were first cultured for 2 days with or without 3F-Neu5Ac (50 μM). Then, medium was changed to osteogenic or adipogenic media with or without 3F-Neu5Ac (50 μM). All subsequent medium changes were without further supplementation of 3F-Neu5Ac, except for the condition labelled “continuous treatment”. Osteogenic medium is standard culture medium supplemented with 0.2 mM ascorbic acid, 0.1 μM dexamethasone, and 20 mM β-glycerophosphate. Adipogenic medium is standard culture medium with 0.5 mM isobutyl methylxanthine, 50 μM indomethacin and 0.5 μM dexamethasone. Matrix mineralization was determined on Day 21 using Alizarin Red S indicator (ARS; Ricca Chemicals, Arlington, TX, USA). Cells were fixed with 10% *v*/*v* formalin solution for 15 min, washed once with PBS, and stained for 20 min with 1% *w*/*v* ARS over gentle shaking. Samples were then washed with PBS, photographed for visual documentation. To quantify ARS staining, wells were incubated with 10% *v*/*v* acetic acid for 30 min, the cell layer scraped, vortexed for 30 s, and centrifuged at 12,000× *g* for 10 min. The optical density of the supernatants was measured at 405 nm. 

For adipogenesis, cells were cultured for 12 days in adipogenic media. Then, cells were fixed with 10% *v*/*v* formalin solution for 15 min, washed once with PBS and stained for 30 min with Oil Red O (Electron Microscopy Sciences, Hatfield, PA, USA). Cells were then washed three times with PBS and incubated with 4% Tween 20 (Affymetrix, Santa Clara, CA, USA) in isopropanol for 5 min, to release the dye. The optical density of supernatants was then measured at 490 nm.

### 4.12. Statistical Analysis

Results are presented as the average with standard error of mean (SEM) as error bars. All experiments were performed at least four times with MSCs derived from different donors, unless otherwise described. Biological replicates for each experiment are denoted by *n* in their respective figure legend. To determine statistical significance, a 1-way ANOVA (followed by post hoc Tukey test) or a paired Student’s *t* test was applied using Graph-Pad Prism Software, depending on the number of conditions to be compared.

## Figures and Tables

**Figure 1 ijms-22-06868-f001:**
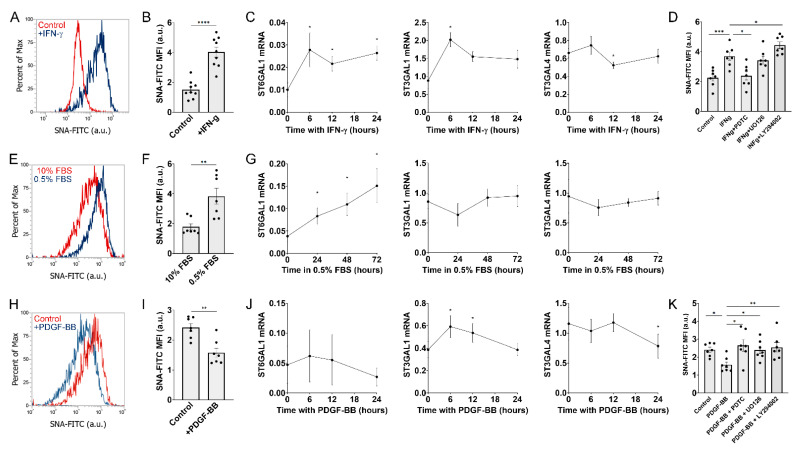
IFN-γ, 0.5%FBS and PDGF-BB modulate sialylations and expression of sialyltransferases. MSCs were cultured for 2 days with either IFN-γ (50 ng/mL), with low fetal bovine serum (0.5% FBS) or PDGF-BB (10 ng/mL) and stained using the lectin SNA, coupled to FITC. (**A**,**E**,**H**) show representative histogram plots, while (**B**,**F**,**I**) show quantification of the mean fluorescent intensities, with each dot representing a biological replicate (i.e., MSCs derived from a different donor). (**C**,**G**,**J**) show mRNA levels of ST6Gal1, ST3Gal1 and ST3Gal4, relative to GAPDH x 1000 (n = 4–5). Differences are calculated in comparison to time 0. (**D**,**K**) show mean fluorescent intensity of cells treated with either IFN-γ or PDGF-BB and specific inhibitors. MFI: Mean Fluorescence Intensity. a.u.: arbitrary units. * *p* < 0.05; ** *p* < 0.005; *** *p* < 0.0005; and **** *p* < 0.00005 as calculated using *t*-test or 1-way ANOVA.

**Figure 2 ijms-22-06868-f002:**
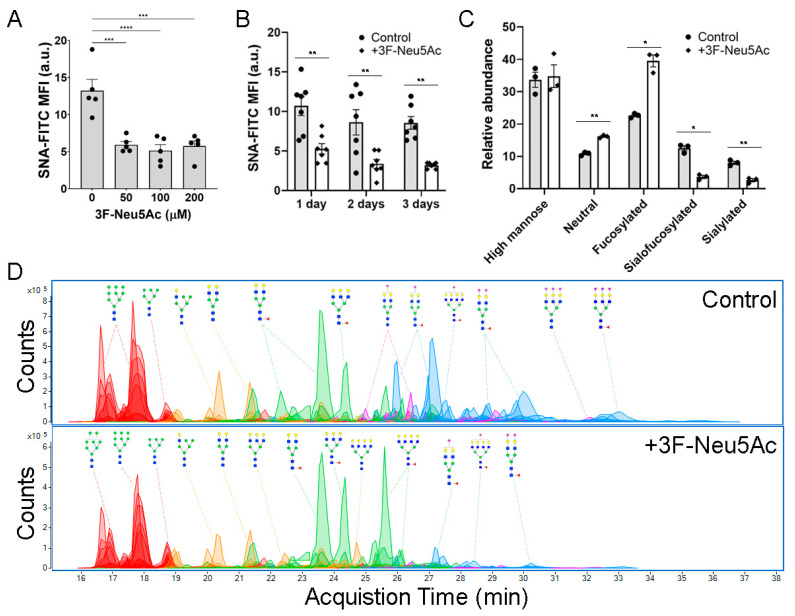
3F-Neu5Ac effectively inhibits sialylation of N-glycans. (**A**) MSCs were cultured for 1 day with varying concentrations of 3F-Neu5Ac and analyzed by flow cytometry using the lectin SNA (n = 5). (**B**) Similarly, MSCs were cultured in the presence of 3F-Neu5Ac (50 μM) for varying amounts of time, prior to analysis by flow cytometry (n = 7). (**C**) MSCs were cultured for 48 h with or without 3F-Neu5Ac (50 μM) and subjected to mass spectrometry to evaluate the impact on different N-glycoforms (n = 3). (**D**) Representative chromatograms of MSCs treated with or without 3F-Neu5Ac, where red peaks show high mannose N-glycans, yellow peaks are neutral N-glycans, green peaks are fucosylated N-glycans, blue peaks are sialofucosylated N-glycans, and purple peaks are sialylated N-glycans. * *p* < 0.05; ** *p* < 0.005; *** *p* < 0.0005; and **** *p* < 0.00005.

**Figure 3 ijms-22-06868-f003:**
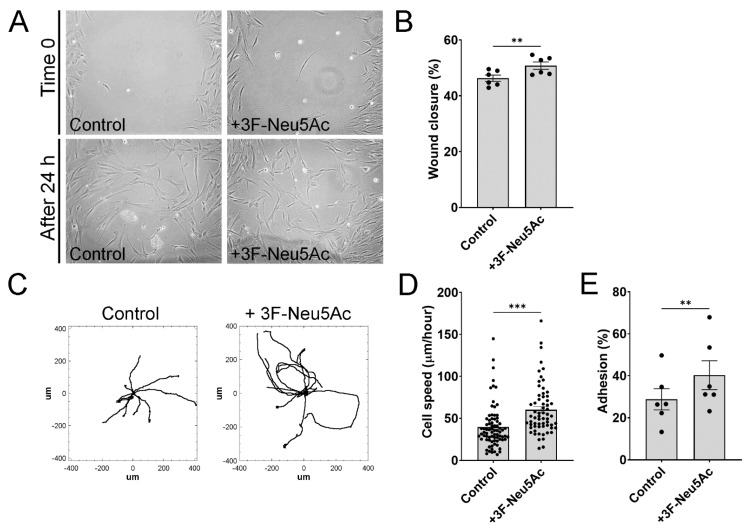
3F-Neu5Ac pre-treatment promotes migration of MSCs in vitro. (**A**) Representative images of wound/scratch assays, immediately after insert removal and 24 h after starting the assay. (**B**) Quantification of wound closure, indicative of higher cell migration (n = 6). (**C**) Representative trajectories of individual cells using videomicroscopy. (**D**) Quantification of cell speed (n = 3). (**E**) MSCs pre-treated with or without 3F-Neu5Ac were lifted to be in suspension and then allowed to attach for 10 min to uncoated culture plates (n = 6). ** *p* < 0.005; *** *p* < 0.0005.

**Figure 4 ijms-22-06868-f004:**
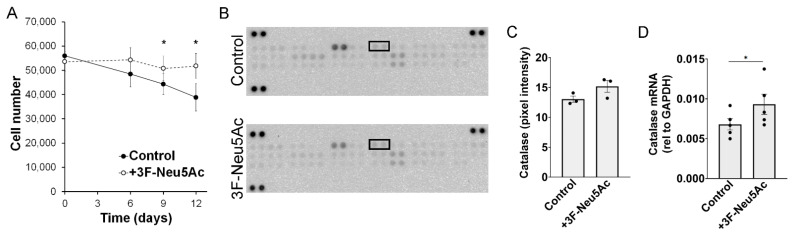
Pre-treatment with 3F-Neu5Ac promotes survival of MSCs in vitro. (**A**) MSCs were cultured for 2 days with or without 3F-Neu5Ac (50 μM). Then, media was changed to serum free media (MEM-α only) and cells transferred to an incubator with 1% oxygen, mimicking an ischemic environment. (**A**) At indicated time points, cells were lifted and counted using Trypan blue exclusion dye and hemocytometer. MSCs pre-treated with 3F-Neu5Ac had significantly higher survival rates after days 9 and 12 in hypoxia (n = 5). (**B**) Representative images of the Apoptosis Proteome Array using lysates from MSCs after 6 days under “ischemia”. Black boxes highlight catalase expression analysis. (**C**) Quantitative representation of pixel intensity for catalase expression on apoptosis array (n = 2). (**D**) 3F-Neu5Ac pre-treatment caused an increase of catalase mRNA expression 6 days after nutrient and oxygen deprivation. Statistical analysis was performed by Student’s *t* test; * = *p* < 0.05.

**Figure 5 ijms-22-06868-f005:**
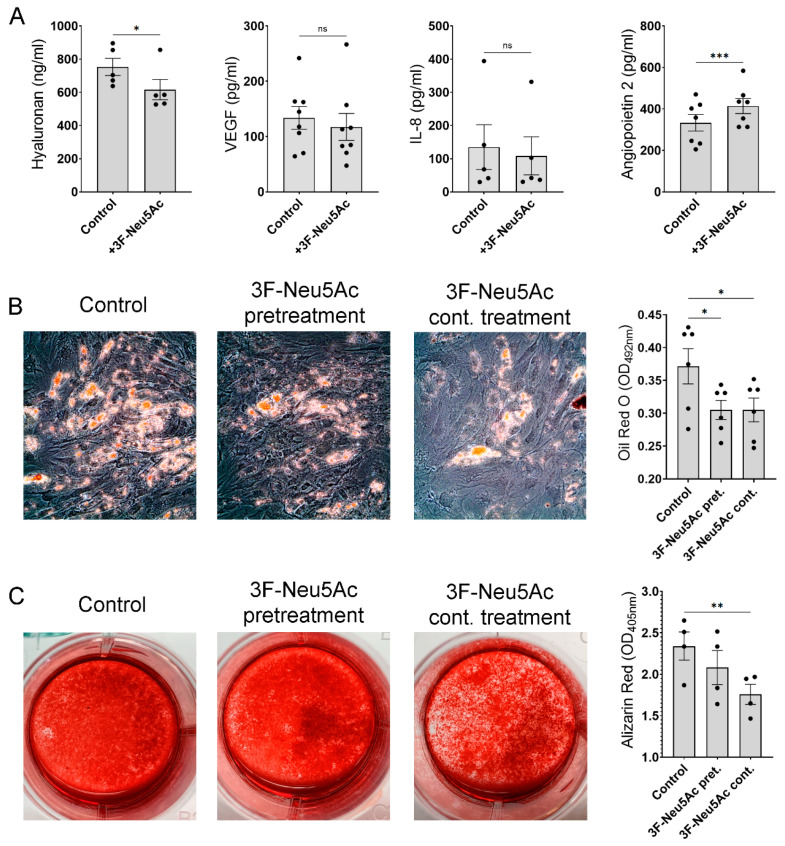
3F-Neu5Ac affects the secretome and differentiation of MSCs. (**A**) MSCs were cultured for 2 days with or without 3F-Neu5Ac (50 μM) and supernatants were collected for determination using ELISA for trophic factors. (**B**) Representative images of MSCs after 12 days in adipogenic media. Adipocytes were stained using Oil Red O (original magnification × 200). Oil Red O staining was subsequently quantified (n = 6). (**C**) Representative images of MSCs after 21 days in osteogenic media. Mineralized calcium is stained with Alizarin Red S. Quantification of Alizarin Red S (n = 4). ns: not significant; * *p* < 0.05; ** *p* < 0.005; and *** *p* < 0.0005.

## Data Availability

All data generated for this manuscript is included in either the main figures or Appendix A. No large datasets (deposited elsewhere) are associated to this work.

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
