# Peer review of "Mesenchymal Stromal Cells Regulate Sialylations of N-Glycans, Affecting Cell Migration and Survival"

_ijms, 2021, doi:10.3390/ijms22136868_

Round 1

Reviewer 1 Report

The paper “Mesenchymal stromal cells regulate sialylations of N-glycans, affecting cell migration and survival” shows how changes in MSCs sialylation could affects the functions/viability of these cells in vitro. The results presented could have important implications in regenerative medicine and other therapeutic approaches. The experiments are well performed and presented efficiently in the paper. The discussion is clear, well written and also, limitations of the current study and further steps are commented. therefore, I recommend the publication of this paper in the current form only with minor modifications:

Line 210: “ealiest” earliest

SNA is a lectin specific for α-2,6sialic acid linkage, for the recognition of α-2,3sialic acids Maackia amurensis lectin should have been used.

Author Response

We thank Reviewer 1 for the careful review and positive feedback. The misspelling on Line 210 was corrected. We also agree that having used SNA has its limitations. According to the manufacturer (Sigma Aldrich): “SNA has an affinity for α-NeuNAc-[2→6]-Gal, α-NeuNAc-[2→6]-GalNAc, and, to a lesser extent, α-NeuNAc-[2→3]-Gal residues”. In accordance, we include the following sentence in our discussion (Lines 280-282): Experiments using other sialic acid-binding lectins are necessary, to further confirm our conclusions and to elucidate if the sialylations attached through specific linkages (e.g. a2,6 vs. a2,3) have differential effects on the cells.   

Reviewer 2 Report

The manuscript by Templeton et al was entitled by “Mesenchymal stromal cells regulate sialylations of N-glycans, affecting cell migration and survival”. They show that the properties of human bone marrow derived mesenchymal stromal cells (MSCs) can be altered by several culturing conditions that affect sialylations of N-glycans such as IFN-gamma, PDGF-BB and low-serum treatments. They also show that supplementation with 3F-Neu5Ac changes the sialylations of N-glycans which correlates with the increased adhesion, migration and survival of MSCs in an in vitro ischemia model. These phenotypic results are interesting. The major weakness of the manuscript is the lack of mechanistic insight of how the altered sialylation of N-glycans would affect the properties of MSCs described. Several minor points are listed as following:

  1. How long were the treatments described in Figure 1D and 1K, 6h, 12h or 24h?
  2. The changes of catalase protein presented in Figure 4B and 4C are inconclusive. Perhaps the authors should consider to use samples 9 or 12 days after 3F-Neu5Ac pretreatment in which they show significant increase of cell survival (Figure 4A). Also, they should increase the n to more than 2 times.
  3. In the conditions of Figure 4, are the changes of sialylation of N-glycans still true 6, 9 and 12 days after pretreatment with 3F-Neu5Ac?

Author Response

We would like to thank Reviewer 2 for the insightful comments. We agree that the underlying mechanism for how altering sialylations impact the cells remains minimally explored. Of note, the underlying mechanism for each of the described observations is likely unique. For example, how reducing sialylations promotes cell migration may not be related to how reducing sialylations promotes survival or affects differentiation. Since this report is to our knowledge a first (and general) description of the role of sialylated N-glycans in MSCs, we did not focus on the exact mechanism for each observed phenomenon. However, we strongly agree that mechanistic insights should be addressed in the future. 

  1. How long were the treatments described in Figure 1D and 1K, 6h, 12h or 24h?

We apologize if this information was not abundantly clear. Under Materials and Methods (lines 349-352), it is stated: MSCs were first treated with the inhibitors and incubated for one hour at 37°C in 20% O2. Then, IFN-g (50 ng/ml) or PDGF-BB (10ng/ml) was added (without a medium change) and cells were incubated for an additional 24 hours, prior to analyzing them using flow cytometry.  

  1. The changes of catalase protein presented in Figure 4B and 4C are inconclusive. Perhaps the authors should consider using samples 9 or 12 days after 3F-Neu5Ac pretreatment in which they show significant increase of cell survival (Figure 4A). Also, they should increase the n to more than 2 times.

We thank Reviewer 2 for this comment. The rationale to focus on day 6, is that the pre-treatment with 3F-Neu5Ac would prevent apoptosis, which is a trend already observable at day 6. In addition, as pointed out below, the effect of 3F-Neu5Ac is expected to be transitory, so we speculated that earlier time points would show greater differences than later time points.

Although the Apoptosis array was meant to only “screen” for potential targets, we acknowledge that showing the results with MSCs derived from only 2 donors seems feeble. We therefore followed the reviewer’s suggestion and repeated this assay with MSCs derived from one additional donor (for a total of 3 biological replicates). Figure 4 was modified accordingly.  

  1. In the conditions of Figure 4, are the changes of sialylation of N-glycans still true 6, 9 and 12 days after pretreatment with 3F-Neu5Ac?

This is a good question that we have not yet addressed. It is also a valid question for the differentiation assays, where pre-treatment with 3F-Neu5Ac inhibited adipogenesis (measured after 12 days). Of note, 0.5% FBS (and most likely serum-free media, as used in the cell survival assays) causes an increase in sialylations. Therefore, the dynamics for how long sialylations remain reduced when treated with 3F-Neu5Ac, will be context dependent. In consequence, we included the following sentence in the Discussion (lines 317-319): “Finally, the effect of 3F-Neu5Ac is expected to be transitory and dependent on protein turnover, but the exact time for how long sialylations remain reduced in response to this inhibitor, remains to be studied.”

Reviewer 3 Report

The manuscript “Mesenchymal stromal cells regulate sialylations of N-glycans, affecting cell migration and survival” from Templeton and co-authors presents an original study, which examined the sialyation of mesenchymal stromal cells isolated from human bone marrow and the effects of a sialyltransferase inhibitor 3F-Neu5Ac (2,4,7,8,9-pentaacetyl-3Fax-Neu5Ac-CO2Me) on a variety of functional cellular responses. The results are well-documented and discussed. It is important to note that the authors recognize that their ‘conclusions are quite contrary to observations in cancer cells’, which would require further investigations in the future and in the context of the role of sialylated N-glycans in cells. There are few minor points to consider for the clarity of the report:

-line 328: specify the commercial source of the human bone marrow; the reference [31] is not appropriate in this case;

- specify the source of 3F-Neu5Ac;

- revise Data Availability Statement.

Author Response

We appreciate the positive evaluation of Reviewer 3 and hope to have addressed comments below:

-line 328: specify the commercial source of the human bone marrow; the reference [31] is not appropriate in this case;

The reviewer is correct that the reference was inappropriate, and we apologize. The reference was removed and the source of bone marrow (Stem Express) was added to the text. By attending to this comment, we realized that the city, state and country for some manufacturers had been inadvertently omitted. They have now all been addressed. Thank you.

- specify the source of 3F-Neu5Ac;

Thank you for noticing this. The manufacturer has now been included (Line 341).

- revise Data Availability Statement.

Thank you for this comment. We included a Data Availability Statement.

Round 2

Reviewer 2 Report

I'm fine with the revisions.